# ChannelNets: Compact and Efficient Convolutional Neural Networks via Channel-Wise Convolutions

**Hongyang Gao**
Texas A&M University
College Station, TX
hongyang.gao@tamu.edu

**Zhengyang Wang**
Texas A&M University
College Station, TX
zhengyang.wang@tamu.edu

**Shuiwang Ji**
Texas A&M University
College Station, TX
sji@tamu.edu

## Abstract

Convolutional neural networks (CNNs) have shown great capability of solving various artificial intelligence tasks. However, the increasing model size has raised challenges in employing them in resource-limited applications. In this work, we propose to compress deep models by using channel-wise convolutions, which replace dense connections among feature maps with sparse ones in CNNs. Based on this novel operation, we build light-weight CNNs known as ChannelNets. ChannelNets use three instances of channel-wise convolutions; namely group channel-wise convolutions, depth-wise separable channel-wise convolutions, and the convolutional classification layer. Compared to prior CNNs designed for mobile devices, ChannelNets achieve a significant reduction in terms of the number of parameters and computational cost without loss in accuracy. Notably, our work represents the first attempt to compress the fully-connected classification layer, which usually accounts for about 25% of total parameters in compact CNNs. Experimental results on the ImageNet dataset demonstrate that ChannelNets achieve consistently better performance compared to prior methods.

## 1 Introduction

Convolutional neural networks (CNNs) have demonstrated great capability of solving visual recognition tasks. Since AlexNet [11] achieved remarkable success on the ImageNet Challenge [3], various deeper and more complicated networks [19, 21, 5] have been proposed to set the performance records. However, the higher accuracy usually comes with an increasing amount of parameters and computational cost. For example, the VGG16 [19] has 128 million parameters and requires $15,300$ million floating point operations (FLOPs) to classify an image. In many real-world applications, predictions need to be performed on resource-limited platforms such as sensors and mobile phones, thereby requiring compact models with higher speed. Model compression aims at exploring a tradeoff between accuracy and efficiency.

Recently, significant progress has been made in the field of model compression [7, 15, 23, 6, 24]. The strategies for building compact and efficient CNNs can be divided into two categories; those are, compressing pre-trained networks or designing new compact architectures that are trained from scratch. Studies in the former category were mostly based on traditional compression techniques such as product quantization [23], pruning [17], hashing [1], Huffman coding [4], and factorization [12, 9].

The second category has already been explored before model compression. Inspired by the Network-In-Network architecture [14], GoogLeNet [21] included the Inception module to build deeper networks without increasing model sizes and computational cost. Through factorizing convolutions, the Inception module was further improved by [22]. The depth-wise separable convolution, proposed in [18], generalized the factorization idea and decomposed the convolution into a depth-wise convolution and a $1 \times 1$ convolution. The operation has been shown to be able to achieve competitive

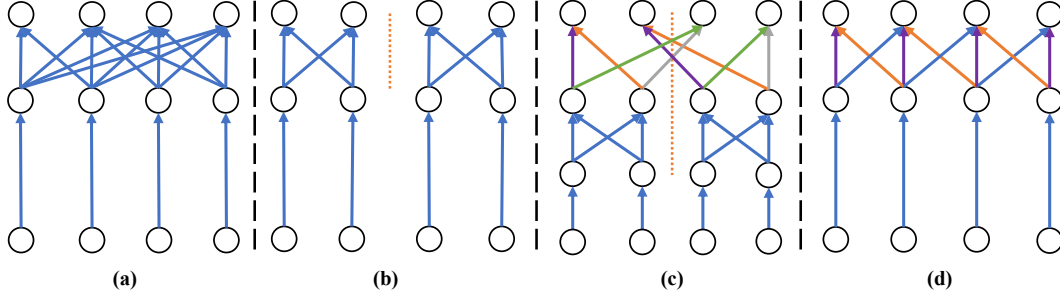

Figure 1: Illustrations of different compact convolutions. Part (a) shows the depth-wise separable convolution, which is composed of a depth-wise convolution and a $1 \times 1$ convolution. Part (b) shows the case where the $1 \times 1$ convolution is replaced by a $1 \times 1$ group convolution. Part (c) illustrates the use of the proposed group channel-wise convolution for information fusion. Part (d) shows the proposed depth-wise separable channel-wise convolution, which consists of a depth-wise convolution and a channel-wise convolution. For channel-wise convolutions in (c) and (d), the same color represents shared weights.

results with fewer parameters. In terms of model compression, MobileNets [6] and ShuffleNets [24] designed CNNs for mobile devices by employing depth-wise separable convolutions.

In this work, we focus on the second category and build a new family of light-weight CNNs known as ChannelNets. By observing that the fully-connected pattern accounts for most parameters in CNNs, we propose channel-wise convolutions, which are used to replace dense connections among feature maps with sparse ones. Early work like LeNet-5 [13] has shown that sparsely-connected networks work well when resources are limited. To apply channel-wise convolutions in model compression, we develop group channel-wise convolutions, depth-wise separable channel-wise convolutions, and the convolutional classification layer. They are used to compress different parts of CNNs, leading to our ChannelNets. ChannelNets achieve a better trade-off between efficiency and accuracy than prior compact CNNs, as demonstrated by experimental results on the ImageNet ILSVRC 2012 dataset. It is worth noting that ChannelNets are the first models that attempt to compress the fully-connected classification layer, which accounts for about 25% of total parameters in compact CNNs.

## 2   Background and Motivations

The trainable layers of CNNs are commonly composed of convolutional layers and fully-connected layers. Most prior studies, such as MobileNets [6] and ShuffleNets [24], focused on compressing convolutional layers, where most parameters and computation lie. To make the discussion concrete, suppose a 2-D convolutional operation takes $m$ feature maps with a spatial size of $d_f \times d_f$ as inputs, and outputs $n$ feature maps of the same spatial size with appropriate padding. $m$ and $n$ are also known as the number of input and output channels, respectively. The convolutional kernel size is $d_k \times d_k$ and the stride is set to 1. Here, without loss of generality, we use square feature maps and convolutional kernels for simplicity. We further assume that there is no bias term in the convolutional operation, as modern CNNs employ the batch normalization [8] with a bias after the convolution. In this case, the number of parameters in the convolution is $d_k \times d_k \times m \times n$ and the computational cost in terms of FLOPs is $d_k \times d_k \times m \times n \times d_f \times d_f$. Since the convolutional kernel is shared for each spatial location, for any pair of input and output feature maps, the connections are sparse and weighted by $d_k \times d_k$ shared parameters. However, the connections among channels follow a fully-connected pattern, *i.e.*, all $m$ input channels are connected to all $n$ output channels, which results in the $m \times n$ term. For deep convolutional layers, $m$ and $n$ are usually large numbers like 512 and 1024, thus $m \times n$ is usually very large.

Based on the above insights, one way to reduce the size and cost of convolutions is to circumvent the multiplication between $d_k \times d_k$ and $m \times n$. MobileNets [6] applied this approach to explore compact deep models for mobile devices. The core operation employed in MobileNets is the depth-wise separable convolution [2], which consists of a depth-wise convolution and a $1 \times 1$ convolution, as illustrated in Figure 1(a). The depth-wise convolution applies a single convolutional kernel independently for each input feature map, thus generating the same number of output channels. The following $1 \times 1$ convolution is used to fuse the information of all output channels using a linear

combination. The depth-wise separable convolution actually decomposes the regular convolution into a depth-wise convolution step and a channel-wise fuse step. Through this decomposition, the number of parameters becomes

$$d_k \times d_k \times m + m \times n, \tag{1}$$

and the computational cost becomes

$$d_k \times d_k \times m \times d_f \times d_f + m \times n \times d_f \times d_f. \tag{2}$$

In both equations, the first term corresponds to the depth-wise convolution and the second term corresponds to the $1 \times 1$ convolution. By decoupling $d_k \times d_k$ and $m \times n$, the amounts of parameters and computations are reduced.

While MobileNets successfully employed depth-wise separable convolutions to perform model compression and achieve competitive results, it is noted that the $m \times n$ term still dominates the number of parameters in the models. As pointed out in [6], $1 \times 1$ convolutions, which lead to the $m \times n$ term, account for 74.59% of total parameters in MobileNets. The analysis of regular convolutions reveals that $m \times n$ comes from the fully-connected pattern, which is also the case in $1 \times 1$ convolutions. To understand this, first consider the special case where $d_f = 1$. Now the inputs are $m$ units as each feature map has only one unit. As the convolutional kernel size is $1 \times 1$, which does not change the spatial size of feature maps, the outputs are also $n$ units. It is clear that the operation between the $m$ input units and the $n$ output units is a fully-connected operation with $m \times n$ parameters. When $d_f > 1$, the fully-connected operation is shared for each spatial location, leading to the $1 \times 1$ convolution. Hence, the $1 \times 1$ convolution actually outputs a linear combination of input feature maps. More importantly, in terms of connections between input and output channels, both the regular convolution and the depth-wise separable convolution follow the fully-connected pattern.

As a result, a better strategy to compress convolutions is to change the dense connection pattern between input and output channels. Based on the depth-wise separable convolution, it is equivalent to circumventing the $1 \times 1$ convolution. A simple method, previously used in AlexNet [11], is the group convolution. Specifically, the $m$ input channels are divided into $g$ mutually exclusive groups. Each group goes through a $1 \times 1$ convolution independently and produces $n/g$ output feature maps. It follows that there are still $n$ output channels in total. For simplicity, suppose both $m$ and $n$ are divisible by $g$. As the $1 \times 1$ convolution for each group requires $1/g^2$ parameters and FLOPs, the total amount after grouping is only $1/g$ as compared to the original $1 \times 1$ convolution. Figure 1(b) describes a $1 \times 1$ group convolution where the number of groups is 2.

However, the grouping operation usually compromises performance because there is no interaction among groups. As a result, information of feature maps in different groups is not combined, as opposed to the original $1 \times 1$ convolution that combines information of all input channels. To address this limitation, ShuffleNet [24] was proposed, where a shuffling layer was employed after the $1 \times 1$ group convolution. Through random permutation, the shuffling layer partly achieves interactions among groups. But any output group accesses only $m/g$ input feature maps and thus collects partial information. Due to this reason, ShuffleNet had to employ a deeper architecture than MobileNets to achieve competitive results.

## 3 Channel-Wise Convolutions and ChannelNets

In this work, we propose channel-wise convolutions in Section 3.1, based on which we build our ChannelNets. In Section 3.2, we apply group channel-wise convolutions to address the information inconsistency problem caused by grouping. Afterwards, we generalize our method in Section 3.3, which leads to a direct replacement of depth-wise separable convolutions in deeper layers. Through analysis of the generalized method, we propose a convolutional classification layer to replace the fully-connected output layer in Section 3.4, which further reduces the amounts of parameters and computations. Finally, Section 3.5 introduces the architecture of our ChannelNets.

### 3.1 Channel-Wise Convolutions

We begin with the definition of channel-wise convolutions in general. As discussed above, the $1 \times 1$ convolution is equivalent to using a shared fully-connected operation to scan every $d_f \times d_f$ locations of input feature maps. A channel-wise convolution employs a shared 1-D convolutional operation, instead of the fully-connected operation. Consequently, the connection pattern between input and

output channels becomes sparse, where each output feature map is connected to a part of input feature maps. To be specific, we again start with the special case where $d_f = 1$. The $m$ input units (feature maps) can be considered as a 1-D feature map of size $m$. Similarly, the output becomes a 1-D feature map of size $n$. Note that both the input and output have only 1 channel. The channel-wise convolution performs a 1-D convolution with appropriate padding to map the $m$ units to the $n$ units. In the cases where $d_f > 1$, the same 1-D convolution is computed for every spatial locations. As a result, the number of parameters in a channel-wise convolution with a kernel size of $d_c$ is simply $d_c$ and the computational cost is $d_c \times n \times d_f \times d_f$. By employing sparse connections, we avoid the $m \times n$ term. Therefore, channel-wise convolutions consume a negligible amount of computations and can be performed efficiently.

## 3.2 Group Channel-Wise Convolutions

We apply channel-wise convolutions to develop a solution to the information inconsistency problem incurred by grouping. After the $1 \times 1$ group convolution, the outputs are $g$ groups, each of which includes $n/g$ feature maps. As illustrated in Figure 1(b), the $g$ groups are computed independently from completely separate groups of input feature maps. To enable interactions among groups, an efficient information fusion layer is needed after the $1 \times 1$ group convolution. The fusion layer is expected to retain the grouping for following group convolutions while allowing each group to collect information from all the groups. Concretely, both inputs and outputs of this layer should be $n$ feature maps that are divided into $g$ groups. Meanwhile, the $n/g$ output channels in any group should be computed from all the $n$ input channels. More importantly, the layer must be compact and efficient; otherwise the advantage of grouping will be compromised.

Based on channel-wise convolutions, we propose the group channel-wise convolution, which serves elegantly as the fusion layer. Given $n$ input feature maps that are divided into $g$ groups, this operation performs $g$ independent channel-wise convolutions. Each channel-wise convolution uses a stride of $g$ and outputs $n/g$ feature maps with appropriate padding. Note that, in order to ensure all $n$ input channels are involved in the computation of any output group of channels, the kernel size of channel-wise convolutions needs to satisfy $d_c \geq g$. The desired outputs of the fusion layer is obtained by concatenating the outputs of these channel-wise convolutions. Figure 1(c) provides an example of using the group channel-wise convolution after the $1 \times 1$ group convolution, which replaces the original $1 \times 1$ convolution.

To see the efficiency of this approach, the number of parameters of the $1 \times 1$ group convolution followed by the group channel-wise convolution is $\frac{m}{g} \times \frac{n}{g} \times g + d_c \times g$, and the computational cost is $\frac{m}{g} \times \frac{n}{g} \times d_f \times d_f \times g + d_c \times \frac{n}{g} \times d_f \times d_f \times g$. Since in most cases we have $d_c \ll m$, our approach requires approximately $1/g$ training parameters and FLOPs, as compared to the second terms in Eqs. 1 and 2.

## 3.3 Depth-Wise Separable Channel-Wise Convolutions

Based on the above descriptions, it is worth noting that there is a special case where the number of groups and the number of input and output channels are equal, *i.e.*, $g = m = n$. A similar scenario resulted in the development of depth-wise convolutions [6, 2]. In this case, there is only one feature map in each group. The $1 \times 1$ group convolution simply scales the convolutional kernels in the depth-wise convolution. As the batch normalization [8] in each layer already involves a scaling term, the $1 \times 1$ group convolution becomes redundant and can be removed. Meanwhile, instead of using $m$ independent channel-wise convolutions with a stride of $m$ as the fusion layer, we apply a single channel-wise convolution with a stride of 1. Due to the removal of the $1 \times 1$ group convolution, the channel-wise convolution directly follows the depth-wise convolution, resulting in the depth-wise separable channel-wise convolution, as illustrated in Figure 1(d).

In essence, the depth-wise separable channel-wise convolution replaces the $1 \times 1$ convolution in the depth-wise separable convolution with the channel-wise convolution. The connections among channels are changed directly from a dense pattern to a sparse one. As a result, the number of parameters is $d_k \times d_k \times m + d_c$, and the cost is $d_k \times d_k \times m \times d_f \times d_f + d_c \times n \times d_f \times d_f$, which saves dramatic amounts of parameters and computations. This layer can be used to directly replace the depth-wise separable convolution.

### 3.4 Convolutional Classification Layer

Most prior model compression methods pay little attention to the very last layer of CNNs, which is a fully-connected layer used to generate classification results. Taking MobileNets on the ImageNet dataset as an example, this layer uses a $1,024$-component feature vector as inputs and produces $1,000$ logits corresponding to $1,000$ classes. Therefore, the number of parameters is $1,024 \times 1,000 \approx 1$ million, which accounts for 24.33% of total parameters as reported in [6]. In this section, we explore a special application of the depth-wise separable channel-wise convolution, proposed in Section 3.3, to reduce the large amount of parameters in the classification layer.

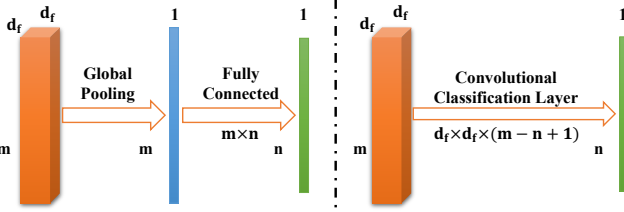

Figure 2: An illustration of the convolutional classification layer. The left part describes the original output layers, *i.e.,* a global average pooling layer and a fully-connected classification layer. The global pooling layer reduces the spatial size $d_f \times d_f$ to $1 \times 1$ while keeping the number of channels. Then the fully-connected classification layer changes the number of channels from $m$ to $n$, where $n$ is the number of classes. The right part illustrates the proposed convolutional classification layer, which performs a single 3-D convolution with a kernel size of $d_f \times d_f \times (m - n + 1)$ and no padding. The convolutional classification layer saves a significant amount of parameters and computation.

We note that the second-to-the-last layer is usually a global average pooling layer, which reduces the spatial size of feature maps to 1. For example, in MobileNets, the global average pooling layer transforms $1,024$ $7 \times 7$ input feature maps into $1,024$ $1 \times 1$ output feature maps, corresponding to the $1,024$-component feature vector fed into the classification layer. In general, suppose the spatial size of input feature maps is $d_f \times d_f$. The global average pooling layer is equivalent to a special depth-wise convolution with a kernel size of $d_f \times d_f$, where the weights in the kernel is fixed to $1/d_f^2$. Meanwhile, the following fully-connected layer can be considered as a $1 \times 1$ convolution as the input feature vector can be viewed as $1 \times 1$ feature maps. Thus, the global average pooling layer followed by the fully-connected classification layer is a special depth-wise convolution followed by a $1 \times 1$ convolution, resulting in a special depth-wise separable convolution.

As the proposed depth-wise separable channel-wise convolution can directly replace the depth-wise separable convolution, we attempt to apply the replacement here. Specifically, the same special depth-wise convolution is employed, but is followed by a channel-wise convolution with a kernel size of $d_c$ whose number of output channels is equal to the number of classes. However, we observe that such an operation can be further combined using a regular 3-D convolution [10].

In particular, the $m$ $d_f \times d_f$ input feature maps can be viewed as a single 3-D feature map with a size of $d_f \times d_f \times m$. The special depth-wise convolution, or equivalently the global average pooling layer, is essentially a 3-D convolution with a kernel size of $d_f \times d_f \times 1$, where the weights in the kernel is fixed to $1/d_f^2$. Moreover, in this view, the channel-wise convolution is a 3-D convolution with a kernel size of $1 \times 1 \times d_c$. These two consecutive 3-D convolutions follow a factorized pattern. As proposed in [22], a $d_k \times d_k$ convolution can be factorized into two consecutive convolutions with kernel sizes of $d_k \times 1$ and $1 \times d_k$, respectively. Based on this factorization, we combine the two 3-D convolutions into a single one with a kernel size of $d_f \times d_f \times d_c$. Suppose there are $n$ classes, to ensure that the number of output channels equals to the number of classes, $d_c$ is set to $(m - n + 1)$ with no padding on the input. This 3-D convolution is used to replace the global average pooling layer followed by the fully-connected layer, serving as a convolutional classification layer.

While the convolutional classification layer dramatically reduces the number of parameters, there is a concern that it may cause a signification loss in performance. In the fully-connected classification layer, each prediction is based on the entire feature vector by taking all features into consideration. In contrast, in the convolutional classification layer, the prediction of each class uses only $(m - n + 1)$ features. However, our experiments show that the weight matrix of the fully-connected classification layer is very sparse, indicating that only a small number of features contribute to the prediction of a class. Meanwhile, our ChannelNets with the convolutional classification layer achieve much better results than other models with similar amounts of parameters.

## 3.5 ChannelNets

With the proposed group channel-wise convolutions, the depth-wise separable channel-wise convolutions, and the convolutional classification layer, we build our ChannelNets. We follow the basic architecture of MobileNets to allow fair comparison and design three ChannelNets with different compression levels. Notably, our proposed methods are orthogonal to the work of MobileNetV2 [16]. Similar to MobileNets, we can apply our methods to MobileNetV2 to further reduce the parameters and computational cost. The details of network architectures are shown in Table 4 in the supplementary material.

**ChannelNet-v1:** To employ the group channel-wise convolutions, we design two basic modules; those are, the group module (GM) and the group channel-wise module (GCWM). They are illustrated in Figure 3. GM simply applies $1 \times 1$ group convolution instead of $1 \times 1$ convolution and adds a residual connection [5]. As analyzed above, GM saves computations but suffers from the information inconsistency problem. GCWM addresses this limitation by inserting a group channel-wise convolution after the second $1 \times 1$

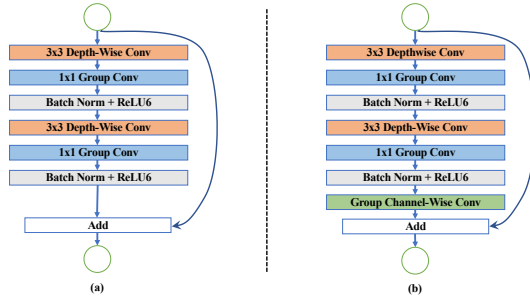

Figure 3: Illustrations of the group module (GM) and the group channel-wise module (GCWM). Part (a) shows GM, which has two depth-wise separable convolutional layers. Note that $1 \times 1$ convolutions is replaced by $1 \times 1$ group convolutions to save computations. A skip connection is added to facilitate model training. GCWM is described in part (b). Compared to GM, it has a group channel-wise convolution to fuse information from different groups.

group convolution to achieve information fusion. Either module can be used to replace two consecutive depth-wise separable convolutional layers in MobileNets. In our ChannelNet-v1, we choose to replace depth-wise separable convolutions with larger numbers of input and output channels. Specifically, six consecutive depth-wise separable convolutional layers with 512 input and output channels are replaced by two GCWMs followed by one GM. In these modules, we set the number of groups to 2. The total number of parameters in ChannelNet-v1 is about 3.7 million.

**ChannelNet-v2:** We apply the depth-wise separable channel-wise convolutions on ChannelNet-v1 to further compress the network. The last depth-wise separable convolutional layer has 512 input channels and 1, 024 output channels. We use the depth-wise separable channel-wise convolution to replace this layer, leading to ChannelNet-v2. The number of parameters reduced by this replacement of a single layer is 1 million, which accounts for about 25% of total parameters in ChannelNet-v1.

**ChannelNet-v3:** We employ the convolutional classification layer on ChannelNet-v2 to obtain ChannelNet-v3. For the ImageNet image classification task, the number of classes is 1, 000, which means the number of parameters in the fully-connected classification layer is $1024 \times 1000 \approx 1$ million. Since the number of parameters for the convolutional classification layer is only $7 \times 7 \times 25 \approx 1$ thousand, ChannelNet-v3 reduces 1 million parameters approximately.

## 4 Experimental Studies

In this section, we evaluate the proposed ChannelNets on the ImageNet ILSVRC 2012 image classification dataset [3], which has served as the benchmark for model compression. We compare different versions of ChannelNets with other compact CNNs. Ablation studies are also conducted to show the effect of group channel-wise convolutions. In addition, we perform an experiment to demonstrate the sparsity of weights in the fully-connected classification layer.

### 4.1 Dataset

The ImageNet ILSVRC 2012 dataset contains 1.2 million training images and 50 thousand validation images. Each image is labeled by one of 1, 000 classes. We follow the same data augmentation process in [5]. Images are scaled to $256 \times 256$. Randomly cropped patches with a size of $224 \times 224$ are used for training. During inference, $224 \times 224$ center crops are fed into the networks. To compare

with other compact CNNs [6, 24], we train our models using training images and report accuracies computed on the validation set, since the labels of test images are not publicly available.

## 4.2 Experimental Setup

We train our ChannelNets using the same settings as those for MobileNets except for a minor change. For depth-wise separable convolutions, we remove the batch normalization and activation function between the depth-wise convolution and the $1 \times 1$ convolution. We observe that it has no influence on the performance while accelerating the training speed. For the proposed GCWMs, the kernel size of group channel-wise convolutions is set to $8$. In depth-wise separable channel-wise convolutions, we set the kernel size to $64$. In the convolutional classification layer, the kernel size of the 3-D convolution is $7 \times 7 \times 25$. All models are trained using the stochastic gradient descent optimizer with a momentum of 0.9 for 80 epochs. The learning rate starts at $0.1$ and decays by $0.1$ at the $45^{th}$, $60^{th}$, $65^{th}$, $70^{th}$, and $75^{th}$ epoch. Dropout [20] with a rate of $0.0001$ is applied after $1 \times 1$ convolutions. We use 4 TITAN Xp GPUs and a batch size of $512$ for training, which takes about 3 days.

## 4.3 Comparison of ChannelNet-v1 with Other Models

We compare ChannelNet-v1 with other CNNs, including regular networks and compact ones, in terms of the top-1 accuracy, the number of parameters and the computational cost in terms of FLOPs. The results are reported in Table 1. We can see that ChannelNet-v1 is the most compact and efficient network, as it achieves the best trade-off between efficiency and accuracy.

We can see that SqueezeNet [7] has the smallest size. However, the speed is even slower than AlexNet and the accuracy is not competitive to other compact CNNs. By replacing depth-wise separable convolutions with GMs and GCWMs, ChannelNet-v1 achieves nearly the same perfor-

Table 1: Comparison between ChannelNet-v1 and other CNNs in terms of the top-1 accuracy on the ImageNet validation set, the number of total parameters, and FLOPs needed for classifying an image.

| Models | Top-1 | Params | FLOPs |
|---|---|---|---|
| GoogleNet | 0.698 | 6.8m | 1550m |
| VGG16 | 0.715 | 128m | 15300m |
| AlexNet | 0.572 | 60m | 720m |
| SqueezeNet | 0.575 | 1.3m | 833m |
| 1.0 MobileNet | 0.706 | 4.2m | 569m |
| ShuffleNet 2x | 0.709 | 5.3m | 524m |
| **ChannelNet-v1** | 0.705 | 3.7m | 407m |

mance as $1.0$ MobileNet with a $11.9\%$ reduction in parameters and a $28.5\%$ reduction in FLOPs. Here, the $1.0$ represents the width multiplier in MobileNets, which is used to control the width of the networks. MobileNets with different width multipliers are compared with ChannelNets under similar compression levels in Section 4.4. ShuffleNet 2x can obtain a slightly better performance. However, it employs a much deeper network architecture, resulting in even more parameters and FLOPs than MobileNets. This is because more layers are required when using shuffling layers to address the information inconsistency problem in $1 \times 1$ group convolutions. Thus, the advantage of using group convolutions is compromised. In contrast, our group channel-wise convolutions can overcome the problem without more layers, as shown by experiments in Section 4.5.

## 4.4 Comparison of ChannelNets with Models Using Width Multipliers

The width multiplier is proposed in [6] to make the network architecture thinner by reducing the number of input and output channels in each layer, thereby increasing the compression level. This approach simply compresses each layer by the same factor. Note that most of parameters lie in deep layers of the model. Hence, reducing widths in shallow layers does not lead to significant compression, but hinders model performance, since it is important to maintain the number of channels in the shallow part of deep models. Our ChannelNets explore a different way to achieve higher compression levels by replacing the deepest layers in CNNs. Remarkably, ChannelNet-v3 is the first compact network that attempts to compress the last layer, *i.e.,* the fully-connected classification layer.

Table 2: Comparison between ChannelNets and other compact CNNs with width multipliers in terms of the top-1 accuracy on the ImageNet validation set, and the number of total parameters. The numbers before the model names represent width multipliers.

| Models | Top-1 | Params |
|---|---|---|
| 0.75 MobileNet | 0.684 | 2.6m |
| 0.75 ChannelNet-v1 | 0.678 | 2.3m |
| **ChannelNet-v2** | **0.695** | 2.7m |
| 0.5 MobileNet | 0.637 | 1.3m |
| 0.5 ChannelNet-v1 | 0.627 | 1.2m |
| **ChannelNet-v3** | **0.667** | 1.7m |

We perform experiments to compare ChannelNet-v2 and ChannelNet-v3 with compact CNNs using width multipliers. The results are shown in Table 2. We apply width multipliers $\{0.75, 0.5\}$ on both MobileNet and ChannelNet-v1 to illustrate the impact of applying width multipliers. In order to make the comparison fair, compact networks with similar compression levels are compared together. Specifically, we compare ChannelNet-v2 with 0.75 MobileNet and 0.75 ChannelNet-v1, since the numbers of total parameters are in the same 2.x million level. For ChannelNet-v3, 0.5 MobileNet and 0.5 ChannelNet-v1 are used for comparison, as all of them contain 1.x million parameters.

We can observe from the results that ChannelNet-v2 outperforms 0.75 MobileNet with an absolute 1.1% gain in accuracy, which demonstrates the effect of our depth-wise separable channel-wise convolutions. In addition, note that using depth-wise separable channel-wise convolutions to replace depth-wise separable convolutions is a more flexible way than applying width multipliers. It only affects one layer, as opposed to all layers in the networks. ChannelNet-v3 has significantly better performance than 0.5 MobileNet by 3% in accuracy. It shows that our convolutional classification layer can retain the accuracy to most extent while increasing the compression level. The results also show that applying width multipliers on ChannelNet-v1 leads to poor performance.

### 4.5  Ablation Study on Group Channel-Wise Convolutions

To demonstrate the effect of our group channel-wise convolutions, we conduct an ablation study on ChannelNet-v1. Based on ChannelNet-v1, we replace the two GCWMs with GMs, thereby removing all group channel-wise convolutions. The model is denoted as ChannelNet-v1(-). It follows exactly the same experimental setup as ChannelNet-v1 to ensure fairness. Table 3 provides comparison results between ChannelNet-v1(-) and ChannelNet-v1. ChannelNet-v1 outperforms ChannelNet-v1(-) by 0.8%, which is significant as ChannelNet-v1 has only 32 more parameters with group

Table 3: Comparison between ChannelNet-v1 and ChannelNet-v1 without group channel-wise convolutions, denoted as ChannelNet-v1(-). The comparison is in terms of the top-1 accuracy on the ImageNet validation set, and the number of total parameters.

| Models | Top-1 | Params |
|---|---|---|
| ChannelNet-v1(-) | 0.697 | 3.7m |
| ChannelNet-v1 | 0.705 | 3.7m |

channel-wise convolutions. Therefore, group channel-wise convolutions are extremely efficient and effective information fusion layers for solving the problem incurred by group convolutions.

### 4.6  Sparsity of Weights in Fully-Connected Classification Layers

In ChannelNet-v3, we replace the fully-connected classification layer with our convolutional classification layer. Each prediction is based on only $(m - n + 1)$ features instead of all $n$ features, which raises a concern of potential loss in performance. To investigate this further, we analyze the weight matrix in the fully-connected classification layer, as shown in Figure 4 in the supplementary material. We take the fully-connected classification layer of ChannelNet-v1 as an example. The analysis shows that the weights are sparsely distributed in the weight matrix, which indicates that each prediction only makes use of a small number of features, even with the fully-connected classification layer. Based on this insight, we propose the convolutional classification layer and ChannelNet-v3. As shown in Section 4.4, ChannelNet-v3 is highly compact and efficient with promising performance.

## 5  Conclusion and Future Work

In this work, we propose channel-wise convolutions to perform model compression by replacing dense connections in deep networks. We build a new family of compact and efficient CNNs, known as ChannelNets, by using three instances of channel-wise convolutions; namely group channel-wise convolutions, depth-wise separable channel-wise convolutions, and the convolutional classification layer. Group channel-wise convolutions are used together with $1 \times 1$ group convolutions as information fusion layers. Depth-wise separable channel-wise convolutions can be directly used to replace depth-wise separable convolutions. The convolutional classification layer is the first attempt in the field of model compression to compress the the fully-connected classification layer. Compared to prior methods, ChannelNets achieve a better trade-off between efficiency and accuracy. The current study evaluates the proposed methods on image classification tasks, but the methods can be applied to other tasks, such as detection and segmentation. We plan to explore these applications in the future.

**Acknowledgments**

This work was supported in part by National Science Foundation grants IIS-1633359 and DBI-1641223.

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
