[Supplementary Material]



# Supplementary Material for "ChannelNets: Compact and Efficient Convolutional Neural Networks via Channel-Wise Convolutions"

Table 4: ChannelNets Architectures. The operations are described in the format of "Type / Stride / # Output channels". "Conv" denotes the regular convolution; "DWSConv" denotes the depth-wise separable convolution; "DWSCWConv" denotes the depth-wise separable channel-wise convolution; "CCL" denotes the convolutional classification layer. The kernel size in regular convolutions and depth-wise convolutions is $3 \times 3$.

| v1 | v2 | v3 |
|---|---|---|
| Conv / 2 / 32 | | |
| DWSConv / 1 / 64 | | |
| DWSConv / 2 / 128 | | |
| DWSConv / 1 / 128 | | |
| DWSConv / 2 / 256 | | |
| DWSConv / 1 / 256 | | |
| DWSConv / 2 / 512 | | |
| GCWM / 1 / 512 | | |
| GCWM / 1 / 512 | | |
| GM / 1 / 512 | | |
| DWSConv / 2 / 1024 | | |
| DWSConv / 1 / 1024 | DWSCWConv | DWSCWConv |
| AvgPool + FC | AvgPool + FC | CCL |

(a)　　　　　　(b)　　　　　　(c)

Figure 4: An example of the weight patterns in the fully-connected classification layer of ChannelNet-v1. Part (a) shows the weight matrix of the fully-connected classification layer. We can see that the weights are sparsely distributed, as most of the weights are in blue color, which indicates zero or near zero values. Part (b) gives a close look of weights in a small region. The histogram in part (c) statistically demonstrates the sparsity of weights.