[Reviews · NeurIPS 2018]

Reviewer 1



The paper proposes channel-wise convolutions that address the full connections between feature maps and replace them with sparse connections (based on 1-D convolutions). This reduces the #params and #FLOPS significantly; while maintaining high accuracy. The authors show results on imagenet classification and compare it to VGG/MobileNet variants to demonstrate this. Strengths: + The paper is well written and easy to follow. Background and related work such as standard convolution+fc layers used in neural nets; mobilenet and shufflenet variants to reduce computation are described in sufficient detail. + The proposed approach is novel and it replaces fully connected pattern of feature maps with 1-D convolutions. As stated, starting from LeNet [13] prior work has experimented with sparse connections between feature maps. In fact libraries like Tensorflow/Torch allow to choose a sparse connection pattern for convolutional layers as well. However, the sparsity in these works is chosen at random and output channels are connected to a fixed fraction of input channels. The proposed work introduces sparsity with 1-D convolutions to convolutional, fully connected and classification layers (i.e. all parts) of a neural net. With appropriate choice of group size and stride, 1-D convolution strategy; it can be ensured that all convolutional channels are used to produce output. + Tab. 1 compares the proposed network to standard approaches and shows good accuracy for less parameters. Tab. 2 does an ablation study on the effects of sparsity on different parts of the network. Convolutional classification causes 3% drop in accuracy but also reduces parameters by ~1M. Weakness: - Results are presented on imagenet classification only. Results on CIFAR or scene classification (Places dataset) or on other tasks such as segmentation/detection will help make the case for channel-wise convolutions stronger. The idea is novel, the paper is well written and results are sufficient to establish that the strategy is effective on imagenet classification. Therefore, I recommend accept.

Reviewer 2



Authors are interested in the problem of reducing the computational cost of large-scale neural networks trained for image recognition. It is a fundamental deep learning problem as the state-of-the-art in image recognition relies on computationally heavy models that require compression to run on resource-limited devices. Based on previously proposed MobileNets, which rely on depth-wise separable convolution, authors propose to replace the fully connected operation used to map the input channels to the outputs channels after depth-wise convolution by a 1-D convolution which drastically reduce the number of parameters. Moreover, they extend this operation to group convolutions and use their model to replace the fully connected classification layer by a convolutional one. Experiments on ImageNet show that the model provides a very good trade-off between accuracy and computational cost (both in terms of parameters and FLOPs), compared to MobileNets and other baselines. Moreover they show that by using width multipliers they can further reduce the number of parameters with a loss in performance which is inferior to comparable baselines. The paper is overall well written but as the vocabulary is heavy (e.g. "depth-wise separable channel-wise convolution) and differences between different types of convolutions can be subtle, the paper could really benefit of making Section 3 easier to follow (more figures). The experiments are convincing and the simplicity of the proposed modifications make this contribution particularly interesting as it could easily be reused in the future.

Reviewer 3



The authors propose an improved version of depth-wise separable convolutions by replacing the fully-connected 1x1 convolutional part with a more sparsely connected layer, which still keeps most of the representational power. In addition, they propose to replace the average pooling layer followed by a fully-connected classifier layer at the end of the network with a single convolutional layer that only uses a small subset of features in the last layer, further reducing the number of parameters. The proposed approach achieves a 10-20% reduction in the number of parameters and FLOPS compared to MobileNet on ImageNet, without a significant decrease in accuracy. The paper is very wordy and convoluted. Although the idea seems relatively simple I did not manage to parse all the details. One thing that seems especially odd to me in how the number of parameters in the classification layer is computed. The authors propose to assign m - n parameters to this layer, where m is the feature dimension of the penultimate layer of the network and n is the number of classes. According to this, a 2-way classifier will require a lot more parameter than a 1000-way, which is counterintuitive. Moreover, if n > m the number of parameters becomes negative. Is there any meaning behind this equation that I'm missing or is it simply incorrect? Overall, the paper seems to contain a non-trivial contribution, but I can't asses it properly due to the unclear presentation and a lack of expertise in the field. After reading the other reviews I'm going to increase my score. I still find the reasoning for selecting the number of parameters in the classification layer provided by the authors in the rebuttal to be speculative, and doubt that the presentation will improve significantly in the camera-ready version, but think the paper can be accepted nevertheless.